# Integrating an Extended-Gate Field-Effect Transistor in Microfluidic Chips for Potentiometric Detection of Creatinine in Urine

**DOI:** 10.3390/s25030779

**Published:** 2025-01-28

**Authors:** Dhaniella Cristhina De Brito Oliveira, Fernando Henrique Marques Costa, Renato Massaroto Beraldo, José Alberto Fracassi da Silva, José Alexandre Diniz

**Affiliations:** Instituto de Química, Universidade Estadual de Campinas (UNICAMP), Campinas 13083-852, SP, Brazil; f231064@dac.unicamp.br (F.H.M.C.); fracassi@unicamp.br (J.A.F.d.S.);

**Keywords:** EGFET, creatinine, sensors, potentiometric detection, 3D printing, UV membrane

## Abstract

Monitoring creatinine levels in urine helps to recognize kidney dysfunction. In this research, we developed a photocurable membrane for the detection of serum creatinine. Using a system based on field-effect transistors, we carried out creatinine quantification in synthetic urine. The device was able to cover values between 3 and 27 mmol L^−1^. The current sensitivity was 0.8529 (mA)^1/2^ mmol^−1^ L with 91.8% linearity, with the LOD and LOQ being 5.3 and 17.5 mmol L^−1^, respectively. The voltage sensitivity was 0.71 mV mmol^−1^ L with a linearity of 96.2%, with the LOD and LOQ being 4.2 and 14.0 mmol L^−1^, respectively. These data were obtained under flow conditions. The system performed very well during the measurements, with a hysteresis of about 1.1%. Up to 90 days after manufacture, the sensor still maintained more than 70% of its initial response. Even when used periodically during the first week and then stored unused at −18 °C, it was able to maintain 96.7% of its initial response. The device used in the flow setup only had a useful life of three days due to membrane saturation, which was not reversible. In the interference test, the membrane was also shown to respond to the urea molecule, but in a different response window, which allowed us to discriminate urea in synthetic urine. EGFETs can be used to identify variations in the creatinine concentration in urine and can help in therapeutic decision-making.

## 1. Introduction

Devices integrated into microfluidic chips allow for the low consumption of reagents and samples, with a simplified recognition process, and the possibility of automation and application in various locations. Among the research into integrated sensors, those that operate by field effect, with detection chips formed from Extended Gate Field-Effect Transistors—EGFETs, have gained special attention. EGFETs are a promising alternative because they are easy to handle and can be used for a wide range of detections due to their extended gate and high input signal sensitivity [1,2].

Potentiometric configurations using EGFETs integrated into microfluidic chips have already been reported in the literature as a strategy to overcome fluid handling complexities, increase recognition sensitivity, and reduce noise of electronic origin [3,4,5,6,7,8]. These devices have mostly been applied to the detection of the hydrogen potential (pH) of different sample media [9,10,11,12,13,14,15,16,17,18]. The operating principle of EGFETs can also be used to build sensors for purposes such as detecting markers of kidney dysfunction.

The prevalence of kidney dysfunction worldwide has shown a worrying annual rise [19]. It is known that in developed countries, at least 10% of the population has some form of renal dysfunction. This does not refer to a specific disease, but to a pathological condition caused by kidney disease or other factors that are not only associated with the kidneys [20].

In situations where the kidney’s filtering and reabsorption functions are mainly compromised by some illness, the tissue chemistry of this organ is altered, which causes variations in glomerular filtration rates (GFRs), which are the main indicator of the clinical stages of kidney dysfunction [21]. GFR estimation is monitored using serum creatinine levels (2-amino-1-methyl-5H-imidazol-4-one), which is one of the end products of protein metabolism in humans and the gold standard for recognizing kidney dysfunction [22].

The main technique for detecting kidney disease is renal tissue biopsy. However, they are usually avoided clinically because they carry a high risk of bleeding [21]. There are also other techniques used to recognize markers of kidney dysfunction, such as Magnetic Resonance Imaging, Magnetic Resonance Spectroscopy, Mass Spectrometry, Spectrophotometry, and others. These techniques are expensive, time-consuming, and some require a lot of sample dilution, which can lead to errors in the measurements [19,20,23].

Due to different factors, less than 10% of people discover that they have kidney dysfunction in the early stages when treatment is the most effective [21]. This is worrying because kidney damage is present in complications of diseases such as Alzheimer’s, Parkinson’s, cancer, diabetes, cardiovascular disease, and multiple sclerosis [23,24]. In medical centers, patients with some type of kidney impairment are also more likely to have a cardiovascular event, multiple organ failures, bone disease, anemia, and uremia, as well as longer hospital stays when undergoing any invasive medical procedures [21,25,26].

In recent years, researchers have proposed non-invasive detection methods, such as the monitoring of metabolites in urine due to its many components that can be used as a possible matrix for the detection of degenerations, mainly renal [23]. Monitoring urine-based markers, when compared to blood-based markers, has some advantages such as non-invasive collection, less influence from circadian rhythm, meal consumption, and stress while also having a longer storage period without significant degradation that can make it impossible to recognize the various clinical species [23].

Creatinine is present in both blood and urine [21,22,27,28,29,30]. As kidney function declines, i.e., there is a risk of death due to the kidneys not being able to perform their function in the body, the creatinine levels in human urine also decrease, indicating that glomerular filtration rates are declining [21]. Although the rate of creatinine in the body is influenced by factors such as gender, body mass, and height, among others, it is estimated that a stable range is between 3 and 27 mmol L^−1^, with a daily urinary decline of between 21% and 31% in these values being indicative of a loss of kidney function [20,31,32]. As well as indicating kidney problems, this marker is also relevant for assessing thyroid function and muscle damage evidenced by the significant decrease in muscle mass in samples that show a creatinine range below 3 mmol L^−1^ [33].

In the literature, a relatively inexpensive approach to creatinine detection has been sensors [33,34,35]. Most of the sensors discussed in the literature, however, assess creatinine concentration from an additional ammonium analysis or by a priori removal of ammonium for subsequent target recognition [27,32]. Directly identifying the concentration of creatinine in biological fluids such as urine therefore presents a research need. In addition, electrochemical biosensors in the literature have used a negative working potential in an attempt to avoid interfering species such as urea and other inorganic salts that coexist with creatinine in urine samples. This low electrode potential makes it impossible to carry out these analyses in real biological fluids [20,27].

Designing a device capable of distinguishing creatinine at a positive electrode potential that is applicable to real biological samples is a task that requires further study. Such scientific investment would enable the population to have rapid access to pathology tests, as well as access to examination responses, and the start of appropriate therapeutic treatments. At the same time, sudden changes in the clinical condition of patients are also factors that require agility in the methods used for recognizing metabolites that are markers of dysfunction. This is especially true if we consider the elderly or premature children, which also call for regular monitoring [36].

Inexpensive, fast-response, and sensitive biosensors can help monitor creatinine levels and are an accurate and reliable strategy for detecting kidney dysfunction. The main objective of this research was to develop a membrane in a gate of the EGFET and integrate it into a microfluidic chip to identify creatinine in urine.

EGFETs have already been reported in the literature and applied in various applications, but never reported in research such as that carried out in this study. The results obtained in the EGFET system were compared with results in the literature and chemical analyses carried out in our laboratory using Capillary Electrophoresis with Capacitively Coupled Contactless Conductivity Detection (C^4^D).

## 2. Materials and Methods

All reagents were of an analytical grade and were used as received, unless otherwise indicated. Hydrochloric acid (HCl), potassium chloride (KCl), and sodium hydroxide (NaOH) from Synth (Diadema, SP, Brazil); Clear V4 acrylic resin from FormLabs (Somerville, MA, United States), and Clear Anycubic resin (Anycubic, Hong Kong, China) were used; L-Histidine (L-His, >99%), 2-(N-morpholino)ethanesulfonic acid (MES, >99%), Potassium tetrakis [3,5- bis(trifluoromethyl)phenyl] borate (KTPB, C_32_H_12_BF_24_K, >95%), buffer solution, synthetic urine, crystallized anhydrous creatinine (98%), deuterated chloroform (CDCl_3_), chloroform, methanol, pyrrole, pure acetone, and powdered graphite were purchased from Sigma Aldrich (Darmstadt, Germany). Mesooctamethylcalix [4]pyrrole (OMCP) was synthesized by our group.

Creatinine samples were prepared in DI water (18.2 MΩ cm water at 25 °C, pH 7.2), in a commercial buffer solution with a pH 4, and in 10 times diluted commercial synthetic urine (pH 5.8). The creatinine concentrations in these matrices were 3, 5, 9, 12, 15, 17, 21, 24 and 27 mmol L^−1^.

### 2.1. Microfluidic Chip

The microfluidic chip was manufactured using a Phrozen Sonic Mini 8 K 3D printer (Phrozen Technology, Taiwan, China). The software used to create the 3D object was Fusion 360 Autodesk (San Rafael, CA, USA). The software used to control the 3D printing and slicing of the object was Chitubox version 1.8 and the resin used for printing was Anycubic Clear (405 nm). The model used had a single-channel structure with sample inlet and outlet, which allowed for the efficient delivery of liquid to the electrodes placed at the side walls of the channel. The design of the chip and its characterization can be found in Appendix A.

The reference electrode used for the tests was an Ag wire coated with AgCl and immersed in 3.0 mol L^−1^ potassium chloride. A detection membrane was made as the indicator electrode, as explained in the following section.

### 2.2. Creatinine Sensing Membrane: Construction and Characterization

To find the best ratio between the recognition species and the selective membrane, we produced a paste using a mixture of Clear V4 acrylic resin and graphite in the proportion 8/9 (*w*/*w*) respectively, of weight in grams of the components as described in a previous work [37] and then added 10 mg of OMCP.

We used fixed proportions of OMCP (10 mg) to variations (*w*/*w*) of conductive paste (50, 100, 200, 300 and 400 mg) using a copper wire as an electrical contact. The mixture was left to cure in a UV projection machine (340–405 nm) for 7 h to harden the membrane. This configuration is referred to throughout the text as Mg1:5, Mg1:10, Mg1:20, Mg1:30 and Mg1:40.

Surface characterization of the sensor membrane was carried out to ascertain its structural properties in order to identify how the composition could affect its functioning. To achieve this, we carried out Attenuated Total Reflectance Fourier Transform Infrared Vibrational Spectroscopy (ATR-FTIR), which helped to obtain information on the molecular configuration and structure of the membrane.

After obtaining information on the best ratio between conductive paste and recognition species, we added KTPB as an ion exchanger in an amount equivalent to 1/3 (in milligrams) of OMCP. Once weighed, the equivalent quantities were mixed in the following order: Clear V4 acrylic resin, OMCP, KTPB, and finally powdered graphite. At each part’s addition, we mixed the components by hand until they were visually well integrated. We then coated the copper wires with the prepared mixture and took them to the photopolymerization machine with 340–405 nm UV projection for 9 h. The Clear V4 acrylic resin was used to enable the membrane to be manufactured in the UV process, the powdered graphite maintained the conductive property of the membrane, and OMCP acted as the recognition molecule, and KTPB as an ion exchanger. Scanning Electron Microscopy was carried out to obtain information on the morphology of the membrane with the addition of each component.

The size of the membrane used in the static measurements was 1 cm covered with 0.51 mm of copper wire. For the flow dynamic measurements, the membrane filled an area of 0.0726 mm.

### 2.3. Commercial MOS Transistor

The commercial model 2N4351 (Motorola, Chicago, IL, USA), which is a device with all four terminals (body, drain, source, and gate) totally isolated, was the commercial MOSFET device chosen for this research.

The electrical performance of the candidate MOSFETs was evaluated. Only commercial MOSFETs that passed the tests and showed similar performance in source–drain current and threshold voltage characteristics were considered for the experiments.

### 2.4. Measurements

Equipment such as voltage power supplies (Minipa MPL 3303M, Shanghai, China), a digital multimeter (Minipa ET-2517 true RMS, Shanghai, China), a syringe pump (Harvard Apparatus, Holliston, MA, USA), a Semiconductor Characterizer System (KI 4200 MPSMU, Keithley, Cleveland, OH, USA), copper wires, and electrical interconnectors were used to carry out the measurements, arranged as shown in Appendix A.

We carried out 2 types of measurements in solution for responses in the triode and saturation region of the MOS. These were (1) variation in the voltage between the gate and source (V_GS_), and between source and drain (V_DS_) measurement, with the working and reference electrodes immersed in solutions contained in a beaker. The data from these measurements are referred to throughout the text as static measurements (SMs). The V_GS_ quoted in the measurements with fixed V_DS_ refers to the potential applied to the reference electrode in relation to the indicator electrode; (2) the same procedure as for measurement 1, with the manufactured microchip and 500 μL min^−1^ flow injected by syringe pump. The data from this measurement are referred to throughout the text as flow measurements (FMs).

In addition to analyzing the source–drain current changes caused by the different concentrations of the solutions, transfer characteristic curves (I_DS_-V_T_) were used as another method of investigating the response to the analytes, analyzing the threshold voltage caused by each concentration of the samples.

### 2.5. Sensor Evaluation Parameters

Source–drain current (I_DS_) and threshold voltage (V_T_) curves as a function of the solution concentration were constructed from the average value of triplicate measurements for each solution concentration. The sensitivity in drain–source current was described as the variation in the linear curve when plotting the square root of I_DS_ in the saturation region versus the concentration of the samples. The sensitivity in voltage was calculated as the variation in the linear curve when plotting V_T_ versus the concentration of the solutions. We used Origin 8.1 to plot the graphs and obtain information from the curves. We used the triple signal-to-noise ratio to calculate the limit of detection (LOD = 3 × Standard error of y − intercept/slope) and limit of quantification (LOQ = 10 × Standard error of y − intercept/slope).

The linear fit of the current and voltage versus concentration curves provided us with the linearity of the measurement. R^2^ values closer to 1 indicated that the measurements were more linear, consequently having a higher percentage when multiplied by 100. We constructed the measurement data curve and calculated the linearity for the different solutions using Origin 8.1.

The sensor’s hysteresis was measured by the I_DS_ variation when the sensor was tested in the initial 15 mmol L^−1^ concentration sequence and then at 3, 5, 9, 12, 17, 21, 24 and 27 mmol L^−1^ of creatinine, where tests were again carried out in the 15 mmol L^−1^ solution during the concentration intervals. The hysteresis calculation used Equation (1) below,(1)$$h=\frac{Vx\left(1\right)-Vx\left(n\right)}{Vx\left(1\right)}100$$
where Vx(1) is the value of the current measured at the first concentration of 15 mmol L^−1^, i.e., at the start of the sequence, and Vx(n) is the value measured at the end of the sequence. Multiplying the value by 100 gives the hysteresis percentage.

For the stability test, the sensor was stored at −18 °C and subsequently tested on the 1st, 3rd, 7th, 75th, and 90th days after manufacture, at creatinine concentrations between 3 mmol L^−1^ and 27 mmol L^−1^.

## 3. Results and Discussions

Initially, to make the creatinine recognition membrane, we varied the ratio of conductive paste, and we set the number of recognition species by 1:40, 1:30, 1:20, 1:10 and 1:5. Figure 1 below shows the behavior of the membranes in the analyte solutions when the concentrations varied logarithmically (p(CH_4_H_7_N_3_O)).

The groups N-H of the OMCP structure attract the oxygen atoms of creatinine to the cavity of the recognition molecule. The charged surface groups form an electrical structure at the membrane–solution interface. A change in the concentration of the solution alters the equilibrium state in the membrane, resulting in a change in electrical properties and surface potential [34].

The ratios 1:40 to 1:10 showed a linear range up to at least a concentration of 10^−3^ mol L^−1^ of creatinine. The Mg1:5 mixture showed the best linearity value in the measurement, with a significant extension of the linear response up to 10^−5^ mol L^−1^ being the greatest of all the proportions tested. After a concentration of 10^−5^ mol L^−1^, there was no significant I_DS_ current variation in the sensor response, as can be seen in Figure 1. The membrane with the lowest amount of conductive paste per fixed amount of OMCP has more interaction points (N-H) with the target species than the other ratios (1:40, 1:30, 1:20, 1:10), which widens the detection window. The results indicated that the ratio between the recognition species and the graphite-based conductive paste influenced the analytical response.

Based on these data, we decided to look at the Attenuated Total Reflection Fourier Transform Infrared Spectrum (ATR-FTIR) for all the membranes. We also carried out this test for the sample with only the graphite-based conductive paste and only with the synthesized OMCP. The spectra are shown in Figure 2a,b below.

For the conductive graphite paste without OMCP, the spectrum in Figure 2 does not show a band in the 3440–3450 cm^−1^ region, which is characteristic of the N-H stretching of OMCP. This also occurs in the 1:40 and 1:30 ratios, where no bands can be seen in this region. At Mg1:20, there is a small N-H band which increases dramatically up to Mg1:5. This could be the cause of the better linear response in the measurements.

In order to obtain high-resolution images for evaluating the membrane’s microstructural characteristics, we used Scanning Electron Microscopy (SEM) as the basic technique.

The following SEM image, Figure 3, shows the surface of the detection membrane throughout its manufacture, demonstrating the different contributions of each component to the morphology of the final membrane.

The photocurable resin, in Figure 3a, showed a homogeneous, smooth surface, without irregularities, with a more compact structure, as is expected for resins used to make objects by 3D printing. When joined with powdered graphite, as demonstrated in Figure 3b, the structure begins to show a structural difference. This shows the morphology of stacked graphite flakes. The flake-like morphology is expected when the proportion of graphite in a mixture is high. In Figure 3c, the addition of OMCP showed significant white staining, demonstrating that the graphite skeleton that remains visible now has particles of the recognition species. Aggregation between the parts, however, does not appear to have been evenly distributed on the membrane surface, with agglomerations at some points. In Figure 3d, the resin with the OMCP and KTPB particles is cured, covering the graphite skeleton, and the graphite flakes become significantly smaller. The aggregation between all the parts is now more visible and the membrane now has a highly irregular surface, with cracks and voids [38,39,40,41,42,43]. The unevenness of the membrane shows that the components are well mixed and that the curing of the resin did not cause the surface to have resin-only characteristics, as shown in Figure 3a. However, some areas of the membrane, which are not reported in Figure 3d, show smoother characteristics. These smoother areas may decrease the sensitivity of the membrane.

Using creatinine concentrations, [C_4_H_7_N_3_O], within the range from 3 mmol L^−1^ to 27 mmol L^−1^ and variation between 21% and 31% in these values, we checked how the I_DS_ current varied for the samples in a pH 4 buffer, water, and synthetic urine using the membrane shown in Figure 3d. Figure 4 below is a compilation of the behavior obtained for the tests in the different matrices tested.

In order to function as an EGFET, the OMCP must bind to the protonated or neutral form of creatinine in the solutions, which leads to charge separation across the membrane interface, culminating in a variation in current and voltage [44,45].

Calix[4]pyrroles species, such as OMCP, are well known in supramolecular chemistry and are used in various types of sensors. They are able to recognize species with different charge properties and present a large number of hosts to act as supramolecular receptors. Modifications to the structure of calix[4]pyrrole, including functionalization with new recognition groups, have increased its bioactivity and stability and open up the possibility of using these species [46,47,48]. Additionally, the OMCP has a well-defined cavity that exhibits charge complementarity with the positively charged creatinine. Its endofunctional cavity ensures greater selectivity for the species. Due to the more displaced nature of the positive charge on the target molecule, its interaction with the OMCP occurs through hydrogen bonding forces and the cooperative effects of CH-π interactions [34,49].

The chemical interaction that occurs between creatinine in solution and the membrane’s OMCP can be briefly explained by the establishment of hydrogen bonds between creatinine’s oxygen and the N-H protons present in the OMCP cavity, which can act as a heteroditopic receptor [34]. Once trapped, the molecule causes a variation in the potential on the surface of the working electrode, which is detected by the MOS through a variation in the device’s I_DS_ current, being that its intensity is proportional to the concentration of creatinine in the sample.

In the attraction between creatinine’s OMCP and the oxygen protons to the cavity of the recognition molecule, the charged surface groups form an electrical structure at the electrode–solution interface. A change in concentration alters the equilibrium state in the membrane, resulting in a change in the electrical properties and surface potential [49].

In the static measurements, the current sensitivity for the measurements was 0.8442 (mA)^1/2^ mmol^−1^ L with a linearity of 92.8% for solutions in water and 0.8390 (mA)^1/2^ mmol^−1^ L with a 90.3% linearity for urine solutions. In a pH 4 buffer, the current sensitivity was 0.9625 (mA)^1/2^ mmol^−1^ L with a linearity of 85.1%. In these measurements, the sensor achieved detection of the minimum concentration tested, which was 3 mmol L^−1^ as shown in Figure 4c,d. For the flow system, the current sensitivity was 2.6819 (mA)^1/2^ mmol^−1^ L with a linearity of 88.2% for the water solutions and 0.8529 (mA)^1/2^ mmol^−1^ L with a 91.8% linearity for synthetic urine solutions. For the pH 4 buffer, the sensor reached 0.9589 (mA)^1/2^ mmol^−1^ L with a linearity of 97.0%, with the minimum detection at a concentration of 3 mmol L^−1^ in all cases.

In Figure 4c, the behavior of the sensitivity values in the current can be explained by the fact that in water (pH 7) there is more of a neutral state of the molecule than in the cation, causing the response to be lower in this matrix than in the buffer (pH 4). In urine solutions, the response decreases because, in this case, the creatinine molecule is not alone in the solution and can be influenced by other species present in the commercial matrix [34,49].

In the flow test, as shown in Figure 4d, the current behavior varies, with creatinine in water having the highest sensitivity of all the samples. This was not expected, since in the pH 4 buffer, due to the molecule being in an acidic environment below its pK_a_ = 4.8, there should be the most protonated state of creatinine and the distribution of the species would be responsible for modulating the charge. What may have occurred is a widening of the source–drain channel for measurements in water, which influenced the analytical response [44,45,49].

The LOD calculated on the basis of the linear fit of the measurements for the water samples in the SM was 4.9 mmol L^−1^ and in the FM was 6.5 mmol L^−1^ for the I_DS_ measurements. For urine, the LODs were 5.5 mmol L^−1^ in SM and 5.3 mmol L^−1^ in FM. For the pH 4 buffer, the LODs were 6.3 mmol L^−1^ in SM and 1.3 mmol L^−1^ in FM. The LOQs were 16.2 mmol L^−1^ in SM and 21.4 mmol L^−1^ in FM for the water solution, and 18.4 mmol L^−1^ in SM and 17.5 mmol L^−1^ in the same test for urine. In the buffer, the LOQs were 20.9 mmol L^−1^ in SM and 4.3 mmol L^−1^ in FM.

Comparing the responses, we see that the greatest linearity occurred for the current measurement of the buffer solution in the flow test. But the greatest sensitivity occurred in the FM water test. The sensitivity values for urine in the flow and static tests were equivalent, while for the water test, the sensitivity varied drastically when the experimental conditions were varied. This variation may have been caused by the possible saturation of the recognition membrane or by the strangulation of the MOS source–drain channel due to the flow of solution in contact with the membrane.

In the voltage sensitivity values highlighted from this point onwards, we must consider that they were much lower than those determined by the Nernst equation, since the variation in the concentrations of the samples was not significant and the concentration values were very close.

Figure 5 shows the detection behavior given by the I_DS_-V_T_ relationship.

The voltage sensitivity for these measurements was 5.63 mV mmol^−1^ L for the solutions in water with a linearity of 89.2%. For the solutions in urine, it was 1.58 mV mmol^−1^ L with a linearity of 95.3%. For the solutions in buffer, it was 2.6 mV mmol^−1^ L with a linearity of 91.0%. The voltage sensitivity for flow measurements was 3.5 mV mmol^−1^ L for the solutions in water with a linearity of 81.9%. For the urine solutions, it was 0.71 mV mmol^−1^ L with a linearity of 96.2%. In the buffer solutions, the values were 2.0 mV mmol^−1^ L with a linearity of 93.8%. In flow, only the samples prepared in urine were able to linearly cover the entire test concentration range. For the water samples, the sensitivity in flow only covered concentrations of 5–24 mmol L^−1^ and, for the buffer, this value was 12–27 mmol L^−1^. In the buffer, it was possible to verify the inversion phenomenon, i.e., when the surface inverts the type of charge that modulates the loading of the MOS input signal [50].

Based on the linear fit of the measurements for the water samples, the LOD for SM was 6.1 mmol L^−1^ and 7.2 mmol L^−1^ for FM. In the urine test, it was 3.9 mmol L^−1^ for SM and 4.2 mmol L^−1^ for FM. In the buffer test, it was 5.2 mmol L^−1^ in SM and 4.9 mmol L^−1^ in FM. The LOQs were 20.4 mmol L^−1^ in SM and 24.1 mmol L^−1^ in FM for water, 13.1 mmol L^−1^ in SM and 14.0 mmol L^−1^ in FM for urine, and 17.2 mmol L^−1^ in SM and 16.2 mmol L^−1^ in FM for the buffer.

In the flow measurements, the voltage sensitivity was higher for the samples prepared in water. For these samples and the urine samples, the variation in the type of test (SM or FM) led to a reduction in sensitivity of around 50% between the static test and the flow test. The actual thickness of the membrane used in the static test was not evaluated by any experimental method and may have been the cause of the divergence in results. However, the voltage sensitivity for the buffer was equivalent for the static and dynamic tests, which already removes the possibility of variation in the result given the divergence in membrane thickness. A comparative table of sensitivity values and other parameters can be found in Appendix A, where it is possible to compare the data with other creatinine detection studies [51,52,53,54,55,56,57,58,59,60,61,62,63,64,65,66,67,68].

The sensor stability results showed a low rate of deviation in the responses to the static measurements in the water tests, 28.2% between the first and last day for the V_T_ test and 5.4% for the I_DS_ values. For the urine tests, the values were 39.8% and 31.0%, respectively. On the other hand, in the buffer, it was 32.0% for V_T_ and 21.3% for I_DS_. From day seven to day seventy-five, the sensor was stored at −18 °C, unused, and under these conditions, it showed a variation in responses of only 3.3%. These data indicated that the greatest losses in the sensor’s responses were related to its use in the measurements.

The deviation rates for the sensor used in the flow measurements were 85.8% for water, 43.7% for urine, and 49.8% for the buffer in the I_DS_ responses, and 39.7%, 92.2%, and 69.1% in the V_T_ responses, respectively, between the first and third day of analysis. Due to the high rates of variation even with storage at low temperatures, the flow measurement configuration was only used in the tests on the first and third day and was discarded for the other measurements on subsequent days.

As for hysteresis, this was only tested for the static measurement device, showing 8.6% variation in the water solutions, 1.1% for the urine tests, and 3.2% for the pH 4 buffer solution. Figure 6 below shows the hysteresis behavior in the different matrices.

As can be seen in Figure 6 above, the hysteresis for the higher concentrations was lower than in the intervals for concentrations below 15 mM (up to test no. 5). This is due to the stability of the measurements for values above the one chosen (15 mM) [34,49].

The selectivity of the sensor was tested with concentrations of urea, [CH_4_N_2_O], since it is one of the predominant species in human urine. According to the literature, urine can contain from 28 mmol L^−1^ to 79 mmol L^−1^ of urea, so we used six points between these initial and final values. In the test, carried out in triplicate, Figure 7 below shows the current and threshold voltage behavior of the membrane in relation to the different concentrations of urea.

The membrane also showed a response to urea, but behaved differently from creatinine detection in terms of the current measurement range and voltage. The current behavior increased with increasing urea concentration instead of a decrease as occurs for creatinine. In addition, the detection range of urea V_T_ is much lower than the values obtained with creatinine, even though there is a large variation in concentration between the respective samples. This means that the two species can be graphically differentiated. However, the selectivity of the membrane, as the data show, is not efficient for detecting creatinine alone.

In tests with interferents that respond in the same window of current and potential variation, an additional electrode with selectivity for the species coexisting with creatinine could be an interesting tool.

The results of the proposed sensor were compared with capillary electrophoresis with contactless conductivity detection and were in agreement (see Appendix A [69]).

## 4. Conclusions

In this research, we sought to develop a device that could measure the variation in creatinine in water, urine, and buffer from static and dynamic measurements in which we investigated the behavior of a field-effect transistor with an extended gate in the detection of this analyte from the responses of variations in the I_DS_ current and in the gate threshold voltage V_T_ of a commercial MOS. The results obtained demonstrated the feasibility of the recognition system. The membrane showed good linearity in the measurements with excellent current and voltage sensitivity values, even in the small concentration variation window.

The sensor used in the static measurements showed a response of over 70% up to at least 90 days after it was made, even though it was used in three different measurements during the first week. This response is even greater when we consider storing the device at low temperatures.

The sensor in the flow system showed good detection results on the first day of testing, but there was a considerable loss of analytical response on the third day of use. This showed the unfeasibility of the flow system for comparative measurements and prolonged use. We believe, however, that this is related to the size of the electrode, which has few points of interaction between the recognition species and the sample. This could be improved with a new microfluidic chip configuration and a new measurement method that considers a smaller amount of flow, or cleaning of the electrode between measurements. As a way of mitigating the rapid loss of response of the membrane and the flow channels, we propose storing the chip at a lower temperature than the one tested (−18 °C) since the variation in this parameter proved to be a possibility for the analytical response. Optimizations in the measurement configuration for future use could also lead to better LOD values for the static measurements. One of the limitations of using the flow test system developed in our study is the need for a sample volume of at least 2 mL to fill the entire flow structure and the test run time.

Overall, the use of EGFETs for creatinine detection has proven to be a versatile, simple, and inexpensive tool that can be adapted to a portable sensor. In the present study, we also present a new way of fabricating the membrane for recognition of this analyte using UV properties, which makes steps such as the immobilization of the recognition species on the transduction surface quickly achievable.

This study advances the state of the art by configuring a new way of developing and composing a membrane selective to serum creatinine. The fabrication of the membrane can also be facilitated by the use of a multi-material 3D printer, since the membrane is developed using a UV process, which also facilitates the fabrication of the microfluidic chip and the selective membrane in a single process.

EGFET sensors could be better discussed with regard to the use of their input signal amplifier function for the determination of molecules such as the one we used in our study. Our sensor also advances the assessment of creatinine concentration without the need for additional analysis of other markers, such as urea, which coexist in real solutions such as urine. The interference of the concentration of this inorganic salt showed a variation in the detection window even with the use of a positive working potential. This is extremely important since the direct identification of creatinine concentration in biological fluids represented a new research need. These are advances that can help in the development of tests on real biological samples.

## Figures and Tables

**Figure 1 sensors-25-00779-f001:**
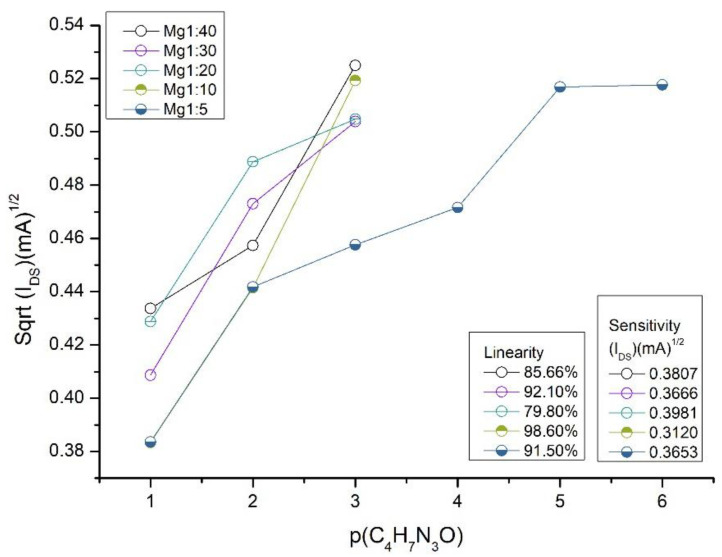
Graph of I_Ds_ signals for different creatinine concentrations when the ratio between graphite-based conductive paste and OMCP was varied. V_GS_ set at 3.5 V and V_DS_ 5 V. Measurements made on 4 mL of sample. Linearity values up to 10^−5^ mol L^−1^ for Mg1:5 and 10^−3^ mol L^−1^ for the other ratios.

**Figure 2 sensors-25-00779-f002:**
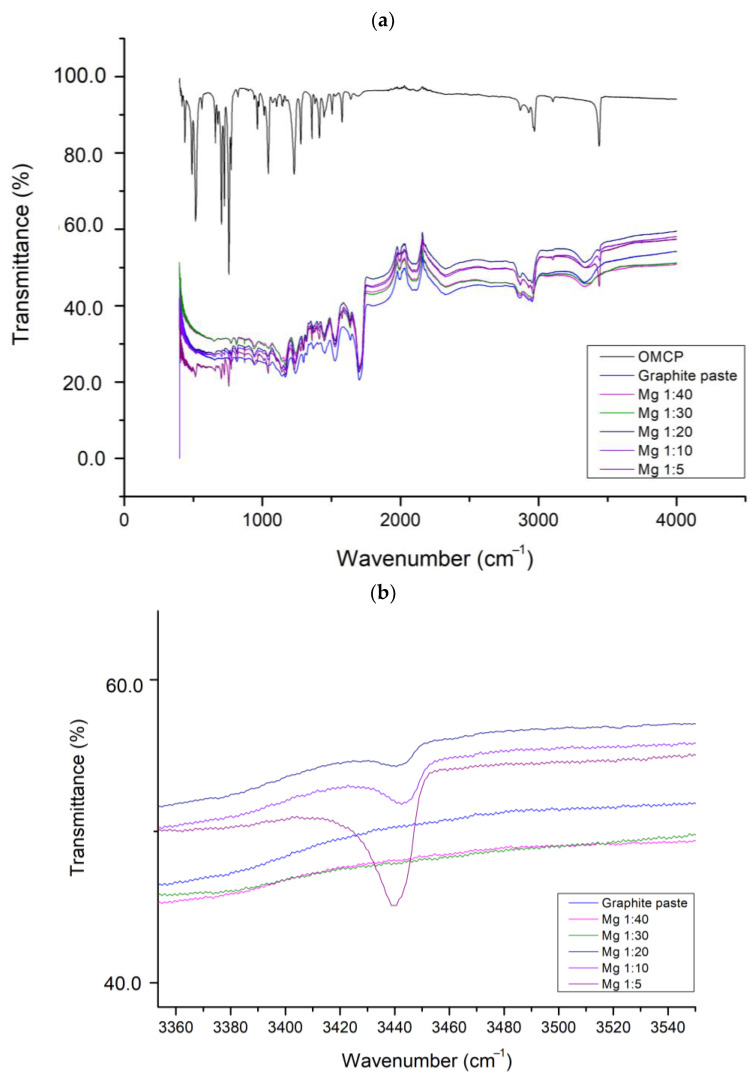
Infrared spectra (ATR-FTIR). (**a**) Set of spectra for different proportions of OMCP + graphite-based conductive paste. (**b**) Zooming the N-H stretching region of the spectrum.

**Figure 3 sensors-25-00779-f003:**
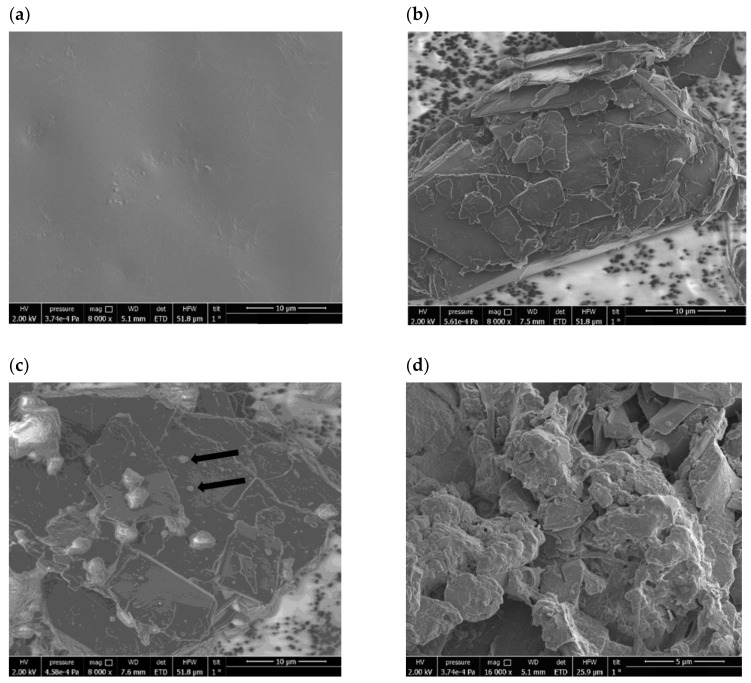
SEM images of the surface of the materials. (**a**) Clear V4 acrylic resin. (**b**) Conductive paste. (**c**) Conductive paste with OMCP. (**d**) Finished membrane.

**Figure 4 sensors-25-00779-f004:**
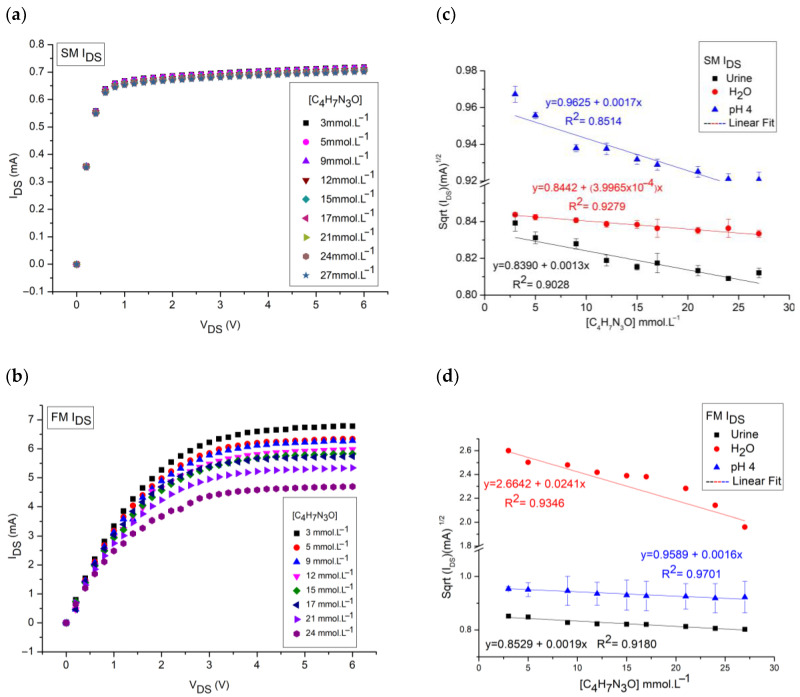
I_DS-_V_DS_ curves. (**a**) SM of creatinine in water, with emphasis on the 5 V_DS_ region. (**b**) FM of creatinine in water. (**c**) Current sensitivity for SM. (**d**) Current sensitivity for FM.

**Figure 5 sensors-25-00779-f005:**
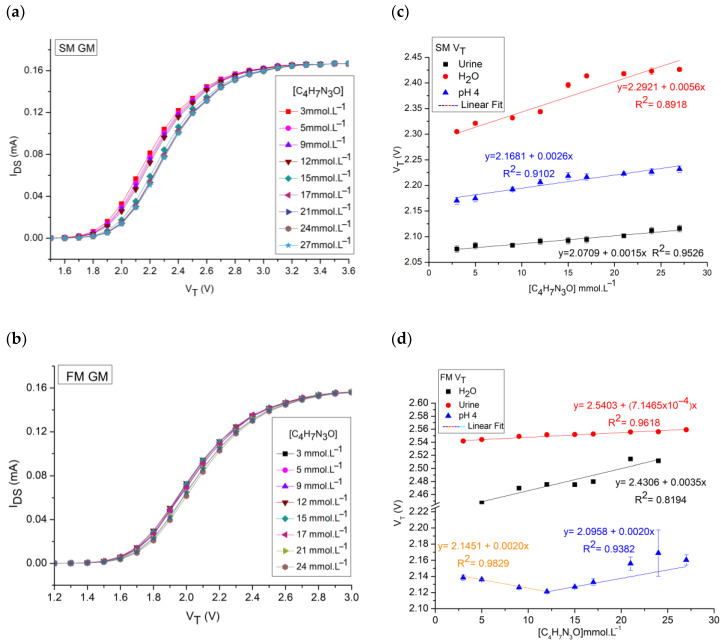
I_DS_-V_T_ curves. (**a**) SM for creatinine in water. (**b**) FM for creatinine in urine. (**c**) Voltage sensitivity for SM. (**d**) Voltage sensitivity for FM.

**Figure 6 sensors-25-00779-f006:**
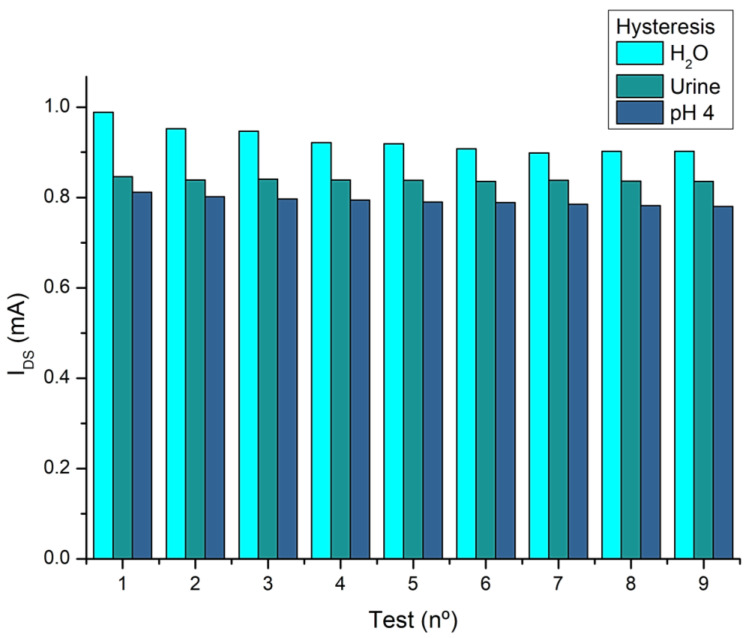
Hysteresis for samples in water, urine, and pH 4 buffer.

**Figure 7 sensors-25-00779-f007:**
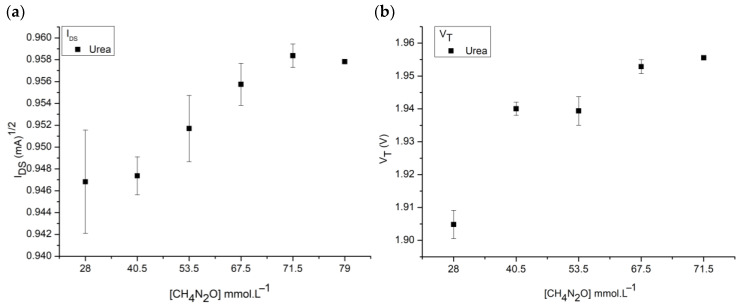
SM selectivity to urea solution (CH_4_N_2_O) in water. (**a**) Current sensitivity. (**b**) Voltage sensitivity.

## Data Availability

The original contributions presented in this study are included in the article/Appendix A. Further inquiries can be directed to the corresponding author.

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
