# Peer review of "Integrating an Extended-Gate Field-Effect Transistor in Microfluidic Chips for Potentiometric Detection of Creatinine in Urine"

_sensors, 2025, doi:10.3390/s25030779_

Round 1

Reviewer 1 Report

Comments and Suggestions for Authors

This article discusses the development of an EGFET-based sensor integrated a microfluidic chip for creatinine detection in urine. The authors optimized the ratio of conductive paste to OMCP for the membrane fabrication of the working electrode. A microfluidic chip was fabricated using a 3D printer. The developed sensors were tested under both static measurement conditions and flow conditions. Their performance in detecting creatinine was evaluated in water, synthetic urine, and pH 4 buffer.

The overall clarity and organization of your manuscript need improvement. Certain sections, such as the methodology, results, and figures, are difficult to follow due to insufficient explanations, unclear distinctions among the results, and low-resolution figures. Upon careful review of this manuscript, several points require revision as follows:

Q1: Since the manuscript focuses solely on testing in a urine environment and does not mention serum, ensure that your title, abstract, and objectives (Line 97-98) align with your experimental design.

Q2: In the abstract, why are there two values of LOQ? How were these values obtained? Please clarify. Additionally, it is not clear which condition the LOD in the abstract refers to.

Q3: The introduction should discuss literature related to OMCP for recognizing species. Do OMCPs recognize other species? Why is OMCP of particular interest for use in this study?

Q4: The importance of the components used in membrane fabrication is unclear. Please explain the function of each component, including Formlab resin, graphite, OMCP, and KTPB. What does 8/9 of graphite refer to?

Q5: Lines 130–136: It is unclear how the ratios were prepared and what is meant by 1:10 and 1:20—are these by weight or volume? Which part is OMCP/graphite/Formlab resin? Please provide detailed information about how the conductive paste was prepared in the methodology.

Q6: What is the difference between OMCP and MCPO? Please recheck all abbreviations to ensure correctness throughout the manuscript.

Q7: How was the electrode fabrication process controlled? For example, what are the dimensions of the working electrode for each ratio? How was reproducibility ensured before testing the performance?

Q8: Line 145: Is the acrylic resin the same as Formlab resin? Please use consistent terminology throughout.

Q9: How were the LOQ values calculated in this study? Please define the calculation method clearly.

Q10: As the authors compared creatinine detection in three solutions (water, pH 4 buffer, and urine), please specify the pH of each solution, especially for synthetic urine.

Q11: Lines 207–209: When stating “We varied the ratio of recognition species to conductive paste by 1:40, 1:30, 1:20, 1:10, and 1:5,” please clarify which value corresponds to the recognition species and which corresponds to the conductive paste. Were these ratios prepared by weight? Please specify.

Q12: In Figure 1, is the x-axis logarithmic? What does p(C4H7N3O) refer to?

Q13: Figures 2(a) and (b) are quite low resolution, and the color lines do not match between the two subfigures. Please provide high-resolution versions with the correct colors that match the legend. Additionally, the number of x-axis (Figure 2b) does not match the context described in Line 233.

Q14: Please discuss why a higher ratio of OMCP leads to a reduction in the N-H bond.

Q15: In Figures 4(a, b) and 5(a, b), what does [C4H2N3O] refer to?

Q16: Line 283: The sentence “In the attraction between creatinine’s OMP and oxygen protons to the cavity of the recognition molecule, the charged surface groups form an electrical structure at the electrode-solution interface” is unclear. Please elaborate.

Q17: Line 295: Please explain how the minimum value of 3 mmol/L was obtained in all cases.

Q18: Please provide a selectivity test for creatinine in comparison to urea and other potential interferences to thoroughly evaluate the performance of the developed sensor in this study.

Q19: In the supplementary table, specify whether the results are from static or flow conditions to facilitate comparison with your work. Clearly state the conditions and sensitivity of each work used for comparison to your study.

Q20: Ensure each mention in the main text specifies the relevant supplementary figure (e.g., "Figure S1") to make it easier for readers to locate the supplementary information.

Author Response

Thank you very much for taking the time to review this manuscript. Please find the detailed responses below and the corresponding revisions/corrections.

We added more information to the manuscript, changed the low-resolution figures and rewrote some paragraphs to better understand our research.

Comments 1: [Q1: Since the manuscript focuses solely on testing in a urine environment and does not mention serum, ensure that your title, abstract, and objectives (Line 97-98) align with your experimental design.]

Response 1: We changed the title of the research, as well as the objective and the data in the abstract, emphasizing the results of the urine test.

Comments 2: [Q2: In the abstract, why are there two values of LOQ? How were these values obtained? Please clarify. Additionally, it is not clear which condition the LOD in the abstract refers to.]

Response 2: There are two LOD and LOQ values because one refers to the IDS test and the other to the VT test. Line: 8-11 and 196-198.

Comments 3: [Q3: The introduction should discuss literature related to OMCP for recognizing species. Do OMCPs recognize other species? Why is OMCP of particular interest for use in this study?]

Response 3: Information about the OMCP can be found in the results section. Its properties that made it suitable for use in this work are described in lines 288-293.

Comments 4: [Q4: The importance of the components used in membrane fabrication is unclear. Please explain the function of each component, including Formlab resin, graphite, OMCP, and KTPB. What does 8/9 of graphite refer to?]

Response 4: Thank you very much for the note. We have changed the text of lines 132-135 for better understanding.

Comments 5: [Q5: Lines 130–136: It is unclear how the ratios were prepared and what is meant by 1:10 and 1:20—are these by weight or volume? Which part is OMCP/graphite/Formlab resin? Please provide detailed information about how the conductive paste was prepared in the methodology.]

Response 5: We have changed the text to highlight the amount in milligrams of each component. Please, see line 135-140. More information on the conductive paste is given in the reference on line 135.

Comments 6:  Q6: What is the difference between OMCP and MCPO? Please recheck all abbreviations to ensure correctness throughout the manuscript.

Response 6: I'm sorry. OMCP and MCPO is the same thing. We rechek all abbreviations.

Comments 7: [Q7: How was the electrode fabrication process controlled? For example, what are the dimensions of the working electrode for each ratio? How was reproducibility ensured before testing the performance?]

Response 7: The size of the membrane used in the static measurements was 1 cm covered with 0.51 mm copper wire. For the flow dynamic measurements, the membrane filled an área of 0.0726 mm. The reproducibility of the electrode was measured by manufacturing in triplicate. In addition, the same paste was used to manufacture both the static and dynamic electrodes.

Comments 8: [Q8: Line 145: Is the acrylic resin the same as Formlab resin? Please use consistent terminology throughout.]

Response 8: Yes. It’s the same. We now use only the commercial name of the resin (Clear V4 acrylic resin) when referring to it

Comments 9: [Q9: How were the LOQ values calculated in this study? Please define the calculation method clearly.]

Response 9: The 3σ and 10σ methods were used to calculate the limit of detection (LOD) and limit of quantification (LOQ) respectively. LOD= 3 x Standard error 196 of y - intercept / slope and LOQ= 10 x Standard error of y - intercept / slope. Line 196-198.

Comments 10: [Q10: As the authors compared creatinine detection in three solutions (water, pH 4 buffer, and urine), please specify the pH of each solution, especially for synthetic urine.]

Response 10: Line 116-119. Creatinine samples were prepared in DI water (18.2 MΩ cm water at 25 ºC, pH 7.2), in commercial buffer solution with pH 4, and in 10 times diluted commercial synthetic 117 urine (pH 5.8).

Comments 11: [Q11: Lines 207–209: When stating “We varied the ratio of recognition species to conductive paste by 1:40, 1:30, 1:20, 1:10, and 1:5,” please clarify which value corresponds to the recognition species and which corresponds to the conductive paste. Were these ratios prepared by weight? Please specify.]

Response 11: We used fixed proportions of OMCP (10mg) to variations (w/w) of conductive paste 136 (50, 100, 200, 300 and 400 mg). This configuration is referred to throughout the text as Mg1:5, Mg1:10, 139 Mg1:20, Mg1:30 and Mg1:40, respectively. Line 136-140.

Comments 12: [Q12: In Figure 1, is the x-axis logarithmic? What does p(C4H7N3O) refer to?]

Response 12: Yes. Its is logarithmic. p(C4H7N3O) is logarithmic creatinine concentrations.

Comments 13: [Q13: Figures 2(a) and (b) are quite low resolution, and the color lines do not match between the two subfigures. Please provide high-resolution versions with the correct colors that match the legend. Additionally, the number of x-axis (Figure 2b) does not match the context described in Line 233.]

Response 13: We replace the image with a higher resolution one and change the text on the line indicated.

Comments 14: [Q14: Please discuss why a higher ratio of OMCP leads to a reduction in the N-H bond.]

Response 14: The greater the amount of conductive paste in the mixture, the less OCMP there will be per area and, as a result, the characteristic peaks of N-H stretching are not favorably visualized.

Comments 15: [Q15: In Figures 4(a, b) and 5(a, b), what does [C4H2N3O] refer to?]

Response 15: [C4H7N3O] refer to creatinine concentrations. Line 278

Comments 16: [Q16: Line 283: The sentence “In the attraction between creatinine’s OMP and oxygen protons to the cavity of the recognition molecule, the charged surface groups form an electrical structure at the electrode-solution interface” is unclear. Please elaborate.]

Response 16: We rewrote it.

Comments 17: [Q17: Line 295: Please explain how the minimum value of 3 mmol/L was obtained in all cases.]

Response 17: In the experimental tests, the sensor responded at the minimum concentration tested, which was 3 mmol L-1 , as can be seen in Figures 4c and 4d.

Comments 18: [Q18: Please provide a selectivity test for creatinine in comparison to urea and other potential interferences to thoroughly evaluate the performance of the developed sensor in this study.]

Response 18: I'm very sorry. That wasn't possible. The 10-day period for amending the manuscript is not sufficient for new tests.

Comments 19: [Q19: In the supplementary table, specify whether the results are from static or flow conditions to facilitate comparison with your work. Clearly state the conditions and sensitivity of each work used for comparison to your study.]

Response 19: We did it.

Comments 20: [Q20: Ensure each mention in the main text specifies the relevant supplementary figure (e.g., "Figure S1") to make it easier for readers to locate the supplementary information.]

Response 20: Thank you for your comment. We've changed all the references to the supplementary material.

Reviewer 2 Report

Comments and Suggestions for Authors

The study presents a groundbreaking integration of EGFETs into microfluidic chips for creatinine detection, showcasing significant potential in non-invasive diagnostics. This research introduces a novel approach to integrating EGFETs into microfluidic systems for potentiometric detection of serum creatinine.  The use of EGFETs in a microfluidic setup is a novel approach for creatinine detection, addressing key challenges in non-invasive diagnostics.The integration of a photocurable membrane in the sensor design is a significant advancement, enabling rapid fabrication and application in real biological samples. Comprehensive experimental design, including static and flow measurements, provides robust data supporting the performance of the EGFET sensors. Detailed characterization of the sensor membrane using techniques like ATR-FTIR and SEM adds depth to the study. The ability to distinguish creatinine from urea and maintain sensitivity across various matrices (water, synthetic urine, and buffer) highlights the practical applicability of the device. Results suggest potential for portable sensor development, expanding access to rapid and cost-effective diagnostics. While some limitations require further optimization, the innovation and clinical relevance of this work are commendable. With additional refinements and validations, this technology could play a pivotal role in advancing point-of-care diagnostics. In this work is a significant contribution to the field and is suitable for journal of Sensors and this manuscript is very well organized. 

Line 31-41, introduction, please re-check the references format. However, the references format in the context appears to follow the general structure for academic citation, but I will highlight areas where improvements or consistency might be needed based on grammar and formatting norms.

Line 135, creatine sensing membrane: construction and Characterization. The UV curing should be provided the condition of wavelength and intensity (UV dose) for fabrication.

Line 210, Results and discussion, In figure 2. The FTIR spectra should label the identification of peaks corresponding to the N-H bond compared to the C-H bond. The factors influencing the thickness of the membrane layer that affect the linearity of signal detection should be explained.

Line 242, Results and discussion. In Fig 3, SEM images could include annotations or enlarged areas for better interpretation of critical features.   

Line 266, Results and discussion. In Fig 4. The factors affecting the sensitivity of creatinine detection in different media should be explained, and it should be demonstrated why the sensitivity in urine is lower than another media. If possible, the pH should be controlled to be consistent lower than pH 4.8 (pKa), and it should be analyzed whether the results are similar across different media.

Line 353-356, Results and discussion. In term of the membrane performance concern, the static system shows good long-term stability, the flow system's rapid degradation (by the third day) raises concerns about practical application. Further discussion on strategies to mitigate this, such as improved electrode design or membrane rejuvenation methods, would strengthen the manuscript.    

Line 371-391, Results and discussion. In term of the selectivity noncerns, the membrane's response to urea, though distinguishable, indicates non-specific interactions that could affect accuracy. The authors should discuss potential strategies for enhancing selectivity, such as molecular imprinting or additional recognition layers.    

Line 392-418, in the conclustion. Although the study compares its findings with prior research, a dedicated section summarizing how this work advances the state-of-the-art would add value. The conclusion effectively summarizes the study but could better emphasize the broader implications, such as scaling for clinical settings or integration with wearable devices.     - Suggestions for future research, particularly addressing the limitations of flow measurements, would be valuable.  

Suggestions for Language and Formatting

In abstract consider rephrasing to enhance clarity and impact. Ensure consistent formatting and up-to-date citations. Cross-check the numbering and alignment with in-text references.

Author Response

Thank you very much for taking the time to review this manuscript. Please find the detailed responses below and the corresponding revisions/corrections.

We added more information to the manuscript, changed the low-resolution figures and rewrote some paragraphs to better understand our research.

Comments 1: [Line 31-41, introduction, please re-check the references format. However, the references format in the context appears to follow the general structure for academic citation, but I will highlight areas where improvements or consistency might be needed based on grammar and formatting norms.]

Response 1: We realized our mistake of not putting the superscript in the first references that we used in the following references. In this way, we have standardized all the indications of references in the text as well as their formatting at the end of the file.

Comments 2: [Line 135, creatine sensing membrane: construction and Characterization. The UV curing should be provided the condition of wavelength and intensity (UV dose) for fabrication.]

Response 2: The mixture was left to cure in a UV projection machine (340-405 nm) for 7 hours to harden the membrane. Line 137-139 and 152. We have no information from the manufacturer about the dosage of the machine.

Comments 3: [Line 210, Results and discussion, In figure 2. The FTIR spectra should label the identification of peaks corresponding to the N-H bond compared to the C-H bond. The factors influencing the thickness of the membrane layer that affect the linearity of signal detection should be explained.]

Response 3: We altered this image in response to another reviewer's recommendation and the image became sharper and more directed towards the N-H stretching peak, which is relevant to this work.

Comments 4: [Line 242, Results and discussion. In Fig 3, SEM images could include annotations or enlarged areas for better interpretation of critical features.] 

Response 4: We added an arrow to the location of the picture we wanted to highlight. There is no significant highlighting on the other images.

Comments 5: [Line 266, Results and discussion. In Fig 4. The factors affecting the sensitivity of creatinine detection in different media should be explained, and it should be demonstrated why the sensitivity in urine is lower than another media. If possible, the pH should be controlled to be consistent lower than pH 4.8 (pKa), and it should be analyzed whether the results are similar across different media.]

Response 5: Thank you for your note. Explanations can be found on lines 320-332.

Comments 6: [Line 353-356, Results and discussion. In term of the membrane performance concern, the static system shows good long-term stability, the flow system's rapid degradation (by the third day) raises concerns about practical application. Further discussion on strategies to mitigate this, such as improved electrode design or membrane rejuvenation methods, would strengthen the manuscript.]

Response 6: We believe, however, that this is related to the size of the electrode, which has few points of interaction between the recognition species and the sample. This could be improved with a new microfluidic chip configuration and a new measurement method that considers a smaller amount of flow or cleaning of the electrode between measurements. As a way of mitigating the rapid loss of response of the membrane of the flow channels we consider storing the chip at a lower temperature than the one tested (-18 ºC) since the variation in this parameter proved to be a possibility for the analytical response. Optimizations in the measurement configuration for future use could also lead to better LOD values for static measurements. One of the limitations of using the flow test system developed in our study is the need for a sample volume of at least 2 mL to fill the entire flow structure and the test run time. Line 439-448.

 Comments 7: [Line 371-391, Results and discussion. In term of the selectivity noncerns, the membrane's response to urea, though distinguishable, indicates non-specific interactions that could affect accuracy. The authors should discuss potential strategies for enhancing selectivity, such as molecular imprinting or additional recognition layers.]    

Response 7: In tests with interferents that respond in the same window of current and potential variation, an additional electrode with selectivity for the species coexisting with creatinine could be an interesting tool. Line 417-418.

Comments 8: [Line 392-418, in the conclustion. Although the study compares its findings with prior research, a dedicated section summarizing how this work advances the state-of-the-art would add value. The conclusion effectively summarizes the study but could better emphasize the broader implications, such as scaling for clinical settings or integration with wearable devices.     - Suggestions for future research, particularly addressing the limitations of flow measurements, would be valuable.]

Response 8: One of the limitations of using the flow test system developed in our study is the need for a sample volume of at least 2 mL to fill the entire flow structure and the test run time. Overall, the use of EGFETs for creatinine detection has proven to be a versatile, simple and inexpensive tool that can be adapted to a portable sensor. In the present study, we also present a new way of fabricating the membrane for recognition of this analyte using UV properties, which makes steps such as the immobilization of the recognition species on the transduction surface quickly achievable. This study advances the state of the art by configuring a new way of developing and composing a membrane selective to serum creatinine. The fabrication of the membrane can also be facilitated by the use of a multi-material 3D printer, since the membrane is developed using a UV process, which also facilitates the fabrication of the microfluidic chip and the selective membrane in a single process. Line 447-458.

Reviewer 3 Report

Comments and Suggestions for Authors

Determination of creatinine is an important diagnostic parameter when assessing kidney damage or when preparing for tomographic studies. Therefore, the creation of new tools for express testing for this marker is an important and urgent task.

This article is quite interesting, but requires a number of important clarifications:

1. In the introduction, the authors should touch upon the issue of using FET-like biosensor structures to detect proteins and creatinine in particular. It is necessary to clearly highlight the advantages of this detection method compared to traditional laboratory methods and other biosensor technologies.

2. In the materials and methods, we would like to see illustrations of the structure and topology of the microfluidic chip, characterization of its channels and a photo of the final structure. Since microfluidics is in the title of the article, it is rather strange to put it in the supplementary files.

3. Also, the article does not contain a diagram of the biosensor structure based on a transistor. The text description of the structure should be illustrated by a diagram understandable to the reader, where the operating principle of the device will be clear. And here I am not talking about the measurement scheme that the authors have, but specifically about the developed sensor structure.

4. In Figure 2, it would be good to see marks for characteristic peak zones and what they correspond to.

5. When studying the selectivity of the developed sensor, why were urine samples with excess of other compounds not studied? Can the membrane respond to pathological conditions associated with excess concentrations of other biomarkers?

6. In conclusion, it would be good to see a table with the final parameters of the biosensor and its comparison with analogues. It would also be good to discuss the introduction of the device into clinical practice.

Author Response

Thank you very much for taking the time to review this manuscript. Please find the detailed responses below and the corresponding revisions/corrections.

We added more information to the manuscript, changed the low-resolution figures and rewrote some paragraphs to better understand our research.

Comments 1: In the introduction, the authors should touch upon the issue of using FET-like biosensor structures to detect proteins and creatinine in particular. It is necessary to clearly highlight the advantages of this detection method compared to traditional laboratory methods and other biosensor technologies.

Response 1: We apologize, but due to the size of the section it was not possible to make the suggested change.

Comments 2: In the materials and methods, we would like to see illustrations of the structure and topology of the microfluidic chip, characterization of its channels and a photo of the final structure. Since microfluidics is in the title of the article, it is rather strange to put it in the supplementary files.

Response 2: Together the authors decided to keep the figure in the supplementary material. We would like to emphasize other points in our work and so we made this choice. In addition, other publications from the group can help to better understand gaps that this article alone cannot solve.

Comments 3: Also, the article does not contain a diagram of the biosensor structure based on a transistor. The text description of the structure should be illustrated by a diagram understandable to the reader, where the operating principle of the device will be clear. And here I am not talking about the measurement scheme that the authors have, but specifically about the developed sensor structure.

Response 3: We apologize. We cross-referenced the comments of the advisors and paid attention to those that were indicated in common. However, due to the sum of the changes requested in the text, we were unable to develop the suggested figure.

Comments 4: In Figure 2, it would be good to see marks for characteristic peak zones and what they correspond to.

Response 4: Figure 2 was changed to another figure with better resolution and the N-H stretching peak, which is so relevant to our research, was better highlighted in Figure 2b.

Comments 5: When studying the selectivity of the developed sensor, why were urine samples with excess of other compounds not studied? Can the membrane respond to pathological conditions associated with excess concentrations of other biomarkers?

Response 5: We only considered urea as an interferent since it is one of the predominant components in our analyte in question. In the future, we plan to carry out other tests to see if other molecules and ions present in the urine interfere. At the moment, we're focusing on the species that is most often discussed in the literature as interfering with creatinine detection.

Comments 6: In conclusion, it would be good to see a table with the final parameters of the biosensor and its comparison with analogues. It would also be good to discuss the introduction of the device into clinical practice.

Response 6: A table is available in the supplementary material. In it, we discuss various creatinine identification devices, considering those that are closest to our proposal. However, working with EGFET for this purpose is a new approach. The research is still underway and the introduction of the sensor in clinical practice, as it currently stands, cannot be considered. We hope that with the changes we are aiming to make to the device, it will be able to meet this alternative.

Round 2

Reviewer 1 Report

Comments and Suggestions for Authors

Thank you for submitting the revised version of your manuscript and for your thoughtful and thorough responses to the reviewers' comments. The revisions are generally satisfactory, and I have some further comments.

- Please recheck all figures, especially Figure 5 and its legend, ensuring they are provided with high-resolution and consistent font sizes.

- The caption for Figure 7 should be modified to identify (a) and (b).

- CHâ‚„Nâ‚‚O should be properly identified and described in the context, not just shown in the figure.

Author Response

Thank you for pointing this out.

Comments 1: [Please recheck all figures, especially Figure 5 and its legend, ensuring they are provided with high-resolution and consistent font sizes.]

Response 1: We checked all the figures. We increased the size of the information contained in the pictures and changed them to images with better resolutions. It should be noted that some figures may continue with a poor resolution, as in the case of figures 4 and 5, due to their size in the text. In the article, the reader will be able to see the image enlarged and the modified resolution will allow them to see the data clearly.

Comments 2: [The caption for Figure 7 should be modified to identify (a) and (b).]

Response 2: We apologize for this lack of information. This has been remedied in the new file we sent. Line 398.

Comments 3: [CHâ‚„Nâ‚‚O should be properly identified and described in the context, not just shown in the figure.]

Response 3: We have identified this on line 392 and in the new legend in figure 7.

Reviewer 3 Report

Comments and Suggestions for Authors

The authors have completed the manuscript well enough that it can be published in the journal in its current form.

Author Response

Thank you for your work in correcting our article. Thank you very much.